# Concatenative Contrastive Sampling for Transformer-based Sequential Recommendation

## Abstract

Sequential recommendation represents a significant research direction in recommender systems, which aims to analyze users' sequential actions to forecast the subsequent item or item sequence they are likely to engage with. This entails deploying machine learning models such as Markov Chains (MC), recurrent neural networks (RNNs), and transformers to unravel the underlying user history patterns in recommender systems and generate recommendations according to their capability in processing sequential data. However, most prior endeavors, while successfully leveraging user history attributes, are constrained in capturing the interplay between user history and new items, as well as the contrastive signals between authentic and unfavorable items. To surmount these limitations, we introduce an attention-based sequential recommendation model with a concatenate-then-split structure that intentionally integrates these interactions. Experimental findings underscore the efficacy of integrating such interactions, with our new model achieving state-of-the-art performance across prevalent sequential recommendation benchmarks.

## 1 Introduction

Recommendation systems play a crucial role in the advancement of e-commerce and content browsing, with wide-ranging applications in various real-world scenarios like online shopping, music streaming, and news reading (Chen et al., 2019a; Wang et al., 2020a; 2022). These systems not only greatly simplify users' content discovery process but also drive higher profits for the online platforms. While traditional recommendation systems, such as Collaborative Filtering (CF) methods (Sarwar et al., 2001), perform reasonably well by assuming static user preferences, recent research has highlighted that user preferences evolve over time. Consequently, incorporating the temporal dynamics of user preferences can lead to more accurate and relevant recommendations.

Sequential recommendation is a special research direction in recommendation systems that focuses on providing personalized recommendations by considering the temporal order of user interactions. It involves analyzing sequential patterns in user behavior to predict the next item or sequence of items a user is likely to engage with. Previous research works have tried using different models to perform the task, including but not limited to Markov chains (He et al., 2018), convolutional neural networks (CNN) (Yuan et al., 2019), and recurrent neural networks (RNN) (Hidasi et al., 2015). However, these models have limited ability to learn user history and predict the interaction scores between the history and the target items to recommend. Very recently, transformers have become a heated neural network architecture, which shows effectiveness in both language-based machine learning tasks and computer vision tasks (Vaswani et al., 2017). Therefore, it would also be favorable if this model could be successfully migrated to sequential learning.

Some prior works in sequential recommendation, such as Kang and McAuley (2018),Li et al. (2020) Sun et al. (2019), Du et al. (2023), Zhang et al. (2023) have also tried to employ the self-attention mechanism and transformers to capture the relationships between items in a user's history. In previous works, the typical approach involves passing user history information (embeddings) to the transformer and then using a tensor product operation to model the interactions between the history items and the target items to predict interaction scores (Kang and McAuley, 2018; Li et al., 2020). While such designs have shown utility due to

the general capability of transformers on sequential data, most of them either overlook or have limited ability to learn the following important information in identifying sequential patterns and making recommendations.

- **(User History ↔ Target Items)**. The aforementioned designs do not treat the target items to be recommended fairly in comparison to the user history. Specifically, the transformer solely learns features from the user history and ignores the features in the target items, which could be considered as "potential user history" in the future. This limitation prevents a comprehensive understanding of the dynamic relationships between the user's past behavior and potential future interactions with the recommended items.

- **(Ground Truth ↔ Negative Items)**. During both the training and the testing stages, the ground truth item must be selected (recommended) from the negative items, which are either randomly sampled from the dataset or items that users did not engage with after display. This process naturally creates contrastive signals among all the items. However, using only a dot product operation is insufficient to effectively learn these cross-item signals.

In this paper, we argue that incorporating these aspects provide richer insights and improve the effectiveness of sequential recommendation models. Based on the above discussions, we present a new design of transformer-based sequential recommendation models that incorporates the above information. We summarize our contribution as follows

- We propose that prior works using self-attentional blocks to generate recommendations treat user history and the target items unfairly. The interactions between user history and target items, and the contrastive signals between ground truth and negative items are not sufficiently learnt by models in the previous works.

- Based on our new understanding of the above two interactions, we introduce a new attention-based sequential recommendation architecture named CTSRec, which explicitly incorporates the aforementioned missing information into the model design.

- Extensive experimental results and ablation studies verify that the aforementioned information is indeed useful, and our proposed method achieves the state-of-the-art performance on the public benchmarks.

## 2 Related Works

Several lines of works are closely related to our research. We first review sequential recommendation, and then we discuss transformers and transformer-based recommendations.

### 2.1 Sequential Recommendation

Sequential recommendation leverages user history sequence information to provide better recommendations, and thus sequential models such as Markov Chains and RNNs are naturally suitable for such tasks. Rendle et al. (2010) proposed to combine matrix factorization with factorized Markov Chains to make next-basket recommendations. Hidasi et al. (2015) introduced Gated Recurrent Unit (GRU) into sequential recommendation and obtained nice results. Yuan et al. (2019) proposed NextItNet to use CNN layers to increase the long-range dependency in user history. Kang and McAuley (2018) first proposed to use self-attention and transformers in sequential recommendation and introduced SASRec. Li et al. (2020) proposed to take into account actual timestamps into the computation of self-attention and extended SASRec by TiSASRec, while Cen et al. (2020) designed an extraction layer for the users' multi-interest.

Knowledge graphs on item relations are observed to drastically improve the performance of recommenders, with the pioneer work CFKG on general recommendation (Zhang et al., 2018). Several works in sequential recommendation utilized this idea. For example, Wang et al. (2020b) modeled the dynamic meaning of an item by combining both the temporal dynamics and the item relations and proposed Chorus. Wang et al.

(2020a) further enhanced the results in Chorus by modeling the temporal evolution of item relations using Fourier transforms to estimate the temporal decay, which significantly outperforms the existing baselines.

## 2.2 Transformers and Attention-based Recommendation

Transformers and attention mechanisms have shown to be effective in different machine learning tasks, including machine translation, caption generation, and image recognition (Xu et al., 2015; Bahdanau et al., 2015; Vaswani et al., 2017; Dosovitskiy et al., 2021), to name a few. The mechanism behind attentional models is to concentrate the model's attention on relevant parts of the input with respect to the output. Specifically, given the query $\mathbf{Q}$, key $\mathbf{K}$ and value $\mathbf{V}$, the scaled dot-product attention used in transformer (Vaswani et al., 2017) is defined as

$$\text{Attn}(\mathbf{Q}, \mathbf{K}, \mathbf{V}) = \text{softmax}\left(\frac{\mathbf{Q}\mathbf{K}^T}{\sqrt{d}}\right)\mathbf{V} \tag{1}$$

In many cases such as recommendation, it is common to observe that $\mathbf{Q}, \mathbf{K}, \mathbf{V}$ are all derived from event sequences. Transformers are constructed by stacking these attention modules with layer-norms layers and multi-layer perceptrons (MLP).

Apart from the aforementioned works in sequential recommendation, attentional modules have proven to be useful in many recommendation tasks such as click-through rate (CTR) prediction, and ranking tasks. Zhou et al. (2018b) proposed DIN which adaptively assigns weights to item embeddings in user history to predict the CTR. Li et al. (2022) proposed a personalized re-ranking model based on contextualized transformers with both item contexts and user contexts. Chen et al. (2019b) utilizes the transformer encoder to learn item representations of historical behaviors. Zhou et al. (2018a) proposed an attentional framework for user-behavior modeling in recommendation.

## 3 Methodology

In this section, we first introduce the preliminaries on the sequential recommendation task, including the notations and the problem formulation. Then, we describe the model design for our new sequential recommendation model.

Table 1: Notations

| Symb. | Description |
|---|---|
| $\mathcal{U}, \mathcal{I}$ | User and Item Set |
| $S_t^u$ | The item user $u$ interacted at time $t$ |
| $N^u$ | The number of historical actions for user $u$ |
| $S^u$ | Historical action sequence $\{(S_1^u, S_2^u, \cdots, S_{N^u}^u\}$ |
| $\mathbb{N}$ | The set of non-negative integers |
| $k$ | Number of items to rank by the model |
| $n$ | The maximum sequence length |
| $B$ | Number of self-attention blocks |
| $d$ | Latent vector dimension |
| $d_p$ | The dimension of the MLP prediction layer |
| $\mathbf{M}^I$ | Item embedding matrix |
| $\mathbf{M}^P$ | Positional embedding matrix |

## 3.1 Preliminaries

In the setting of sequential recommendation, each user $u \in \mathcal{U}$ has a sequence of historical actions $S^u = \{S_1^u, S_2^u, \cdots, S_{N^u}^u\}$ on the item set $\mathcal{I}$, i.e., $S_t^u \in \mathcal{I}, \forall t \in [1, N^u] \cap \mathbb{N}$, where $N^u$ denotes the number of historical actions made by user $u$. At each time step $t$, the task is to consider all the historical actions before $t$, and make recommendations for the next item or the next series of items to engage with for every user $u$.

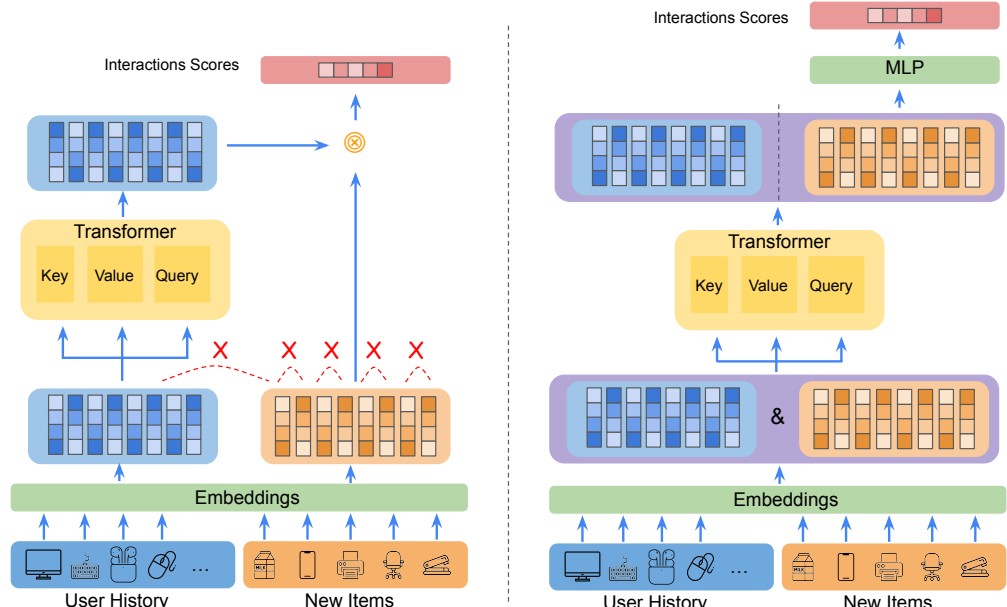

Figure 1: A comparison of the history-item based self-attention model model architecture and our proposed model architecture. *Left:* the history-item based self-attention model model which uses the self-attention layers to learn user history but not the target items. The red cross and dashed lines denote the missing information from (1). user history ↔ target items and (2). ground truth ↔ negative items. *Right:* Our new attention-based sequential recommendation model where the user history and the target item embeddings are concatenated to be the input to the self-attention blocks.

In this work, we study the sequential recommendation under the next item prediction setting: during training, the sequence of input to the machine learning model is $\{S_1^u, S_2^u, \cdots, S_{N^u-1}^u\}$ and the ground truth (label) sequence is $S^u = \{S_2^u, \cdots, S_{N^u}^u\}$. To align with prior works, at each timestep, the model will be given a list of $k$ new items $\mathbf{r} = \{r_1, r_2, \cdots r_k\}$. The $k$ target items contain the ground truth item and $k-1$ negative items, which are either randomly sampled or simply items not chosen by the users. In the most difficult case, $k = |\mathcal{I}|$. The model is required to generate an ordered list of all the $k$ items and the performance of the model is evaluated by the order of the ground truth item (see the Experiments section for the metrics).

### 3.2 Input Processing and Embeddings

Similar to prior works, at any timestep $t \geq 1$, we transform the training historical sequence $(S_1^u, S_2^u, \cdots, S_t^u)$ into a fixed-length sequence $\mathbf{s} = (s_1, s_2, \cdots, s_n)$ as the input (Kang and McAuley, 2018; Li et al., 2020), where $n$ is the maximum sequence length. Only the most recent $n$ items are used if $t \geq n$, and padding items are added to the left of the sequence if $t < n$. The user history sequence $\mathbf{s}$ and the target items $\mathbf{r}$ are then fed into the same item embedding layer $\mathbf{M}^I \in \mathbb{R}^{|\mathcal{I}| \times d}$ to obtain the embedding matrices $E_{his} \in \mathbb{R}^{n \times d}$ and $E_{new} \in \mathbb{R}^{k \times d}$ respectively.

$$E_{his} = \begin{bmatrix} m_{s_1} \\ m_{s_2} \\ \cdots \\ m_{s_n} \end{bmatrix}, E_{new} = \begin{bmatrix} m_{r_1} \\ m_{r_2} \\ \cdots \\ m_{r_k} \end{bmatrix}, \tag{2}$$

where each $m_{s_i} \in \mathbb{R}^{1 \times d}$. Constant zero vectors are used for padding items. Besides, we also create a learnable positional embedding layer $\mathbf{M}^P$ of the items because otherwise self-attentional models would not be aware of the position of the previous items $P \in \mathbb{R}^{n \times d}$.

$$P = \left[ p_1^T, p_2^T, \cdots p_n^T \right]^T \tag{3}$$

The positional embedding is added element-wise to the user history embedding as part of the input to the model.

### 3.3 The New Self-Attentional Model-CTSRec

As we have mentioned in the Introduction section, the interactions between the two embedding matrices $E_{his}$ and $E_{new}$ (user history $\leftrightarrow$ target items), and the interactions among the rows in $E_{new}$ (ground truth $\leftrightarrow$ negative items) are of vital importance. Therefore, instead of examining the two embeddings $E_{his}$ and $E_{new}$ separately, in this work we propose to treat them (almost) equally in the attentional model with a concatenate-then-split (CTS) structure. We concatenate the sum of user history embedding $E_{his}$ and positional embedding, with all the target item embedding $E_{new}$, and then use the whole concatenated embedding as the transformer input $\widehat{E} \in \mathbb{R}^{(n+k)\times d}$.

$$\widehat{E} = \begin{bmatrix} E_{his} + P \\ E_{new} \end{bmatrix} \tag{4}$$

#### 3.3.1 Model architecture Overview

We present a overview of our proposal architecture comparison between history-item based self-attention models such as SASRec and TiSASRec(Kang and McAuley, 2018; Li et al., 2020) and our proposed new model is shown in Figure 1. As can be observed from the figure, history-item based self-attention model uses the self-attention block to learn solely user history features (Kang and McAuley, 2018; Li et al., 2020), but leaves the item embeddings out of the learning process. The interactions between user history and item embeddings are merely modeled by a tensor dot product operation, which is highly insufficient. Moreover, the interactions between the items are never modelled since the tensor product operation is not a cross-item operation. Our model, on the other hand, concatenates the user history embeddings and item embeddings together as the input to the self-attention blocks. In this way, the attentions blocks are able to learn from both the user history and the target items at the same time, and utilize the two aforementioned information to obtain better recommendations.

#### 3.3.2 The Self-Attention Block

Our model contains multiple ($B$) self-attentional blocks. In the following, we will discuss the structure of the first block and the other blocks can be inferred. Given the input $\widehat{E}$, we compute the output of the self-attention block $Z$ by the following equation

$$Z = \text{Attn}(\widehat{E}W^Q, \widehat{E}W^K, \widehat{E}W^V)$$
$$= \text{softmax}\left(\frac{\widehat{E}W^Q\left(\widehat{E}W^K\right)^T}{\sqrt{d}}\right)\widehat{E}W^V \tag{5}$$

where the projection matrices $W^Q, W^K, W^V \in \mathbb{R}^{d\times d}$. The scaling factor $1/\sqrt{d}$ is used to standardize the values.

#### 3.3.3 Causality

Note that the user history part of our input to the transformer is sequential, and the target items should never be observed before the last historical action. Therefore to prevent information leakage, the input embeddings for the items corresponding to $(s_1, s_2, \cdots, s_t)$ should be masked when predicting the next item $s_{t+1}$, for every $t \in [1, n-1]$. In addition, the target items $(r_1, r_2, \cdots, r_k)$ should only be observed for items no earlier than $s_n$. Consequentially, to encode the causality relations, we mask out all the attention links between $\mathbf{Q}_i$ and $\mathbf{K}_j$ that satisfy $j > i$ and $1 \le i < n$ where $i, j$ are positive integers in the range $[1, n+k]$ and $\mathbf{Q} = \widehat{E}W^Q$, $\mathbf{K} = \widehat{E}W^K$ similar to casual mask between the history items Vaswani et al. (2017)

### 3.3.4 MLP, LayerNorm, Residual, and Dropout

Similar to prior works (Kang and McAuley, 2018; Li et al., 2020), we have added multi-layer perceptrons (MLP) to the self-attentional block to improve the model with additional nonlinearity. Specifically, we add a two-layer MLP with shared parameters for each row $z_i$ in the self-attention block output $Z$ (See Equation (5))

$$\text{MLP}(z_i) = \max(0, z_i W_1 + b_1) W_2 + b_2 \tag{6}$$

with $W_1, W_2 \in \mathbb{R}^{d \times d}$ and $b_1, b_2 \in \mathbb{R}^{1 \times d}$. We also use Layer normalization (LayerNorm) layers (Ba et al., 2016) and residual connections and dropout to to stablize the neural network architecture and reduce overfitting.

$$\widehat{Z}_i = z_i + \text{Dropout}(\text{MLP}(\text{LayerNorm}(z_i))) \tag{7}$$

The output $\widehat{Z}$, will be as the input to the more blocks of stacked self-attention. To avoid confusion, we denote the output of the $b$-th self-attention block as $\widehat{Z}^{(b)}$.

### 3.3.5 Prediction and Loss Computation

After passing the input embeddings through stacked blocks of self-attentional, we obtain the output vector $Z$ which should be used to make a prediction on the interaction scores. Since $Z \in \mathbb{R}^{(n+k) \times d}$ is used to make predictions on only the $k$ candidate items, we perform output projection to match the shape of the labels. Since before the self-attentional layers, we have used concatenation to obtain the user history embedding and the target items embedding, it would be natural to apply splitting and use the channels (rows) for item embeddings to make the final prediction. In particular, if the output $Z^{(B)}$ is

$$\widehat{Z}^{(B)} = \left[ \widehat{z}_{s_1}^{(B)T}, \cdots \quad \widehat{z}_{s_n}^{(B)T}, \quad \widehat{z}_{r_1}^{(B)T}, \cdots \quad \widehat{z}_{r_k}^{(B)T} \right]^T \tag{8}$$

Then the split output $\overline{Z_t^{(B)}}$ is chosen to be the last $k$ rows.

$$\overline{Z^{(B)}} = \left[ \widehat{z}_{r_1}^{(B)T}, \widehat{z}_{r_2}^{(B)}, \cdots, \widehat{z}_{r_k}^{(B)T} \right]^T \tag{9}$$

$\overline{Z_t^{(B)}}$ is then fed into a final projection layer to predict the interaction scores for all the items for recommendation.

$$\widehat{y}_j = \text{MLP}_p(\widehat{z}_{r_j}^{(B)}) = \max(0, \widehat{z}_{r_j}^{(B)} W_p^{(1)} + b_p^{(1)}) W_p^{(2)} + b_p^{(2)} \tag{10}$$

where the weights of projection includes $W_p^{(1)} \in \mathbb{R}^{d \times d_p}$ and $W_p^{(2)} \in \mathbb{R}^{d_p \times 1}$. The final output $\widehat{y}$ from the prediction layer lies in the real space $\mathbb{R}^{k \times 1}$.

The model output is then used to compute and optimize the Bayesian Personalization Ranking (BPR) loss proposed by Hidasi and Karatzoglou (2018) and extended by Wang et al. (2023), which is used to optimize ranking outcome (Wang et al., 2020b;a; 2022; 2023). Specifically, we have used the multi-ary version of this loss function, by letting the predicted interaction score for the ground truth item being $\widehat{y}_+$. The loss for a single user is computed by

$$\mathcal{L}_{BPR} = -\frac{1}{k} \sum_{j=1}^{k} \log \left( \sigma \left( \widehat{y}_+ - \widehat{y}_j \right) \right) \tag{11}$$

where $\sigma(x) = 1/(1 + e^{-x})$ is the sigmoid function. The loss for a mini-batch is the average BPR loss across the users in the batch.

### 3.4 Complexity Analysis

**Time Complexity** The computational complexity of CTSRec training is dominated by the attention layer and the feed-forward network, resulting in a complexity of $O(n^2 d + nkd + k^2 d + kd^2 + nd^2)$. By further

leveraging sequence parallelization, the computation can be evenly split onto each local token from within the length $n$ event history as well as in $k$ candidate items, resulting in a amortized $O(nd + nkd + kd + d^2)$. The computation cost for inference is similar to that of training.

**Space Complexity** The space complexity of CTSRec is dominated by embeddings and parameters as well as the self-attention layers, and feed-forward networks. The asymptotic total number of parameters is $O(|\mathcal{I}|nd + kd + d^2)$, and $o(|\mathcal{U}|)$ in terms of the number of users.

**Handing Larger** $k$ The efficacy of our method benefits from the extra $k$ term in the computation complexity. While potential efficiency optimization such as importance sampling could be incorporated as further investigations, we empirically demonstrate in our experiments that a adequate amount of computation of $k = 20$ strikes a balance significant performance boost and afforable computation overhead.

## 4 Experiments

Table 2: Basic dataset statistics

| Dataset | ♯ user | ♯ item | avg actions/user | ♯ actions |
|---------|--------|--------|------------------|-----------|
| G & GF  | 14.7K  | 8.5K   | 9.92             | 145.8K    |
| CP & A  | 27.9K  | 10.4K  | 6.97             | 194.4K    |
| Games   | 24.3K  | 10.6K  | 9.54             | 231.8K    |
| ML-1M   | 6.0K   | 3.4K   | 163.5            | 987K      |

In this section, we provide the experimental setup, the results of our proposed method on multiple public benchmarks, and the discussion of the effectiveness of the new model with ablation studies. More comparisons and additional experiments can be found in the Appendix.

### 4.1 Experimental Setup

#### 4.1.1 Implementation Details

We have used the open-sourced ReChorus library (Wang et al., 2020b) for the implementation of all the baseline algorithms and our new recommendation model. The BPR loss with multiple negative samples during training is already supported by the original codebase. All experiments are conducted with a single Nvidia A100 GPU.

#### 4.1.2 Hyper-parameter Details

We have set the maximum sequence length to be $n = 20$, the latent vector dimension $d = 64$, and the dimension of the MLP predictin layer $d_p = 256$. To make a fair comparison, all the models are trained with the same number of negative samples ($k = 99$) during training, except for the ablation study on the number of negative samples in Section 4.3. The models are tested with the standard procedure as in prior works, i.e., 100 items are provided to the model with the ground truth item to be recommended (Li et al., 2020; Wang et al., 2019; 2022).

#### 4.1.3 Baselines

We compare our model with the following baselines. We remark that the above methods are already competitive baselines that represent the state-of-the-art (SOTA) models in the field of sequential recommendation.

- FPMC (Rendle et al., 2010): A recommendation model that combines matrix factorization and factorized first-order Markov Chains.

- GRU4Rec (Hidasi et al., 2015): The first model that uses RNNs in sequential recommendation.

- Caser (Tang and Wang, 2018): A model that embeds the sequence of recent items into an "image" in the time and latent spaces.

- SASRec (Kang and McAuley, 2018): The first model that incorporates the self-attention mechanisms in sequential recommendation.

- TiSASRec (Li et al., 2020): An improved model of SASRec that combines self-attention and time intervals in user history to model user interest.

- ComiRec (Cen et al., 2020): The first model that incorporate the users' multiple interest into the sequential recommendation process.

- ContraRec (Wang et al., 2023): A general model to add context-context contrast signals to sequetial recommendation algorithms. We have followed the original paper to use BERT as the encoder model.

- TimiRec (Wang et al., 2022): A new model that combines time-interval modeling and multi-interest knowledge distillation to further improve the performance of different models including transformers.

Some recent works, such as SLRC (Wang et al., 2019), Chorus (Wang et al., 2020b) and KDA (Wang et al., 2020a), utilize the additional information of the item relations to provide better sequential recommendations. Therefore, it would be unfair to compare our algorithm against these models.

### 4.1.4  Datasets

We present the experimental results on four popular sequential recommendation benchmarks. Summary statistics of these datasets are provided in Table 2.

- **Amazon**. This is a series of e-commerce datasets with reviews of products crawled from Amazon.com. We choose three categories "Grocery and Gourmet Food (G & GF)", "Cell Phones and Accessories (CP&A)" and "Video Games (Games)". The datasets are highly sparse.

- **MovieLens-1M (ML-1M)**. This is a widely-used **dense** benchmark dataset for evaluating sequential recommendation algorithms. The items are different movies, and the user actions are ratings of the movies.

We have followed the same preprocessing procedure of the prior works for all the datasets(Li et al., 2020; Wang et al., 2019; 2020b; 2022; 2023).

### 4.1.5  Evaluation Metrics

To evaluate the quality of the sequential recommendation, we use Hit Ratio (HR) and Normalized Discounted Cumulative Gain (NDCG). Given a user set $\mathcal{U}$, and that $g_u$ denotes the rank of the ground-truth item for user $u$, the mathematical expressions for the two metrics are as follows.

$$
\begin{aligned}
\text{HR@K} &= \frac{1}{|\mathcal{U}|} \sum_{u \in \mathcal{U}} I(g_u \leq K) \\
\text{NDCG@K} &= \frac{1}{|\mathcal{U}|} \sum_{u \in \mathcal{U}} \frac{I(g_u \leq K)}{\log_2(g_u + 1)}
\end{aligned}
\tag{12}
$$

In short, HR@K measures the frequency of the ground-truth item appearing in Top-K recommendations among all users, whereas NDCG@K adds a ranking position weight to the original measure. Both are standard measures in sequential recommendation.

### 4.2 Recommendation Performance

Table 3 and 4 summarize the performance of all baselines and the new model. For ease of comparison and to follow the prior works such as Wang et al. (2020b; 2022), we take K=5 and K=10 in HR@K and NDCG@K in all our experiments. Among all the baseline methods, TimiRec (Wang et al., 2022) achieved the state-of-the-art performance because of its ability to model both time intervals in the recommendation process, and the multi-interest of user history. Although our new model does not have such explicit modeling of the information and has a simple pipleline, as can be observed, our model still out-performs all the existing baselines by a significant margin on all the datasets.

Table 3: Comparisons between the baseline methods and the new method of `HR@K` and `NDCG@K` with `K=5` and `K=10`, on Amazon Grocery and Gourmet Food (G & GF), Cell Phones and Accessories (CP & A). The results are averaged over 10 independent runs. The highest results in each column are highlighted in bold.

| Method | Amazon G & GF | | | | Amazon CP & A | | | |
| --- | --- | --- | --- | --- | --- | --- | --- | --- |
| | K=5 | | K=10 | | K=5 | | K=10 | |
| | HR | NDCG | HR | NDCG | HR | NDCG | HR | NDCG |
| FPMC | 0.362 | 0.283 | 0.443 | 0.309 | 0.400 | 0.302 | 0.508 | 0.336 |
| GRU4Rec | 0.417 | 0.314 | 0.518 | 0.347 | 0.467 | 0.351 | 0.590 | 0.391 |
| Caser | 0.408 | 0.305 | 0.507 | 0.337 | 0.446 | 0.329 | 0.573 | 0.370 |
| TiSASRec | 0.397 | 0.306 | 0.482 | 0.333 | 0.452 | 0.344 | 0.565 | 0.381 |
| ComiRec | 0.375 | 0.270 | 0.476 | 0.302 | 0.440 | 0.328 | 0.555 | 0.366 |
| ContraRec | 0.422 | 0.326 | 0.510 | 0.356 | 0.468 | 0.360 | 0.583 | 0.397 |
| TimiRec | 0.426 | 0.320 | 0.517 | 0.350 | 0.469 | 0.356 | 0.588 | 0.395 |
| **CTSRec (ours)** | **0.433** | **0.331** | **0.526** | **0.372** | **0.482** | **0.362** | **0.610** | **0.403** |

Table 4: Comparisons between the baseline methods and the new method of `HR@K` and `NDCG@K` on Amazon Video Games (Games) and MovieLens-1M. The results are averaged over 10 independent runs. The highest results in each column are highlighted in bold.

| Method | Amazon Games | | | | ML-1M | | | |
| --- | --- | --- | --- | --- | --- | --- | --- | --- |
| | K=5 | | K=10 | | K=5 | | K=10 | |
| | HR | NDCG | HR | NDCG | HR | NDCG | HR | NDCG |
| FPMC | 0.574 | 0.445 | 0.688 | 0.482 | 0.591 | 0.435 | 0.737 | 0.482 |
| GRU4Rec | 0.612 | 0.473 | 0.727 | 0.510 | 0.691 | 0.541 | 0.798 | 0.575 |
| Caser | 0.572 | 0.435 | 0.692 | 0.474 | 0.692 | 0.541 | 0.794 | 0.574 |
| TiSASRec | 0.610 | 0.477 | 0.721 | 0.513 | 0.736 | 0.593 | 0.824 | 0.622 |
| ComiRec | 0.575 | 0.437 | 0.695 | 0.476 | 0.693 | 0.553 | 0.800 | 0.577 |
| ContraRec | 0.617 | 0.486 | 0.728 | 0.522 | 0.723 | 0.589 | 0.811 | 0.603 |
| TimiRec | 0.624 | 0.487 | 0.735 | 0.523 | 0.731 | 0.591 | 0.821 | 0.621 |
| **CTSRec (ours)** | **0.637** | **0.497** | **0.752** | **0.534** | **0.745** | **0.605** | **0.835** | **0.635** |

### 4.3 Ablation Study

In this subsection, we aim at answering the following research questions.

- **(Q1).** Since the user history is split out from the output of the transformer, is it useful in the learning process?

- **(Q2).** How useful is the concatenation operation and to predict the interaction scores together? Is it better to predict the interaction scores one-item-by-one-item?

- **(Q3).** Is the BPR loss function necessary for the good performance of our model?

- **(Q4).** How does the transformer model architecture affect the performance of our model?

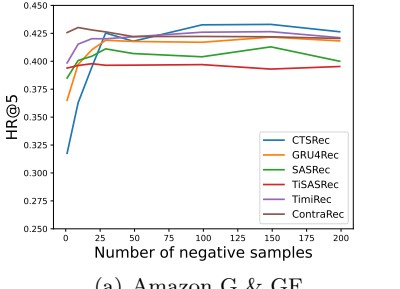
(a) Amazon G & GF

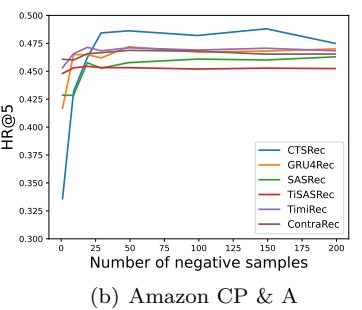
(b) Amazon CP & A

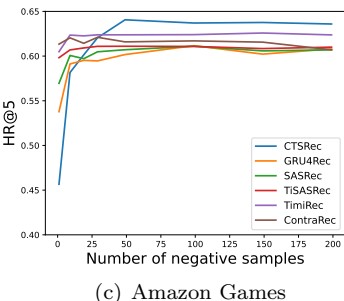
(c) Amazon Games

Figure 2: Ablation study on the number of negative items. In each figure, we take $k - 1 = 1, 9, 19, 29, 49, 99, 149, 199$.

- **(Q5).** How does the number of negative items affect the performance of all the models?

**(Q1)** questions the model's ability in learning the interactions between user history and the target items to recommend, due to the fact that we have removed user history from the output of the transformer for the final prediction by the MLP. **(Q2)** essentially doubts the usefulness of the contrastive signals in the target items and whether learning their interactions with the user history individually can enhance the understanding of each item. Since some prior works use different loss functions (Kang and McAuley, 2018; Li et al., 2020), **(Q3)** questions whether the performance of our model is consistent on the other loss functions. **(Q4)** is related to the neural architecture robustness. **(Q5)** is the most interesting question because the number of negative items affect the strength of the contrastive signals in the training stage. If there are too few negative samples, the model maybe incapable of learning the difference between the ground truth and the negatives. However, the model might also be distracted by the data imbalance if there are too many negative samples. Moreover, it is also interesting to see whether changing the number of negative items will affect other models, especially prior transformer-based models which has limited ability to learn such information.

We have conducted the ablation study on the Amazon G& GF dataset to provide the answers to the above questions. The results are provided in Table 5 and Figure 2.

Table 5: Ablation study (HR@K and NDCG@K) on the Grocery and Gourmet Food dataset. The sign ↓ indicates a significant performance drop compared with the original baseline in Table 3.

| Method | Grocery and Gourmet Food | | | |
|---|---|---|---|---|
| | K=5 | | K=10 | |
| | HR | NDCG | HR | NDCG |
| **baseline** | **0.433** | **0.331** | **0.526** | **0.372** |
| (1) no history ↓ | 0.209 | 0.132 | 0.341 | 0.289 |
| (2) item-by-item ↓ | 0.413 | 0.313 | 0.516 | 0.345 |
| (3) BCE loss | **0.441** | **0.340** | **0.538** | **0.388** |
| (4) separate transformer ↓ | 0.403 | 0.309 | 0.505 | 0.337 |
| (5) 1 block ↓ | 0.402 | 0.303 | 0.499 | 0.322 |
| (6) 2 blocks | 0.416 | 0.321 | 0.515 | 0.341 |
| (7) 3 blocks | 0.429 | 0.319 | 0.518 | 0.355 |
| (8) 5 blocks | 0.431 | 0.329 | 0.523 | 0.359 |

**(1). No history**. To answer **(Q1)**, we have removed user history from the model architecture and trained the model on the item embeddings only. It could be observed that there is a significant drop in the HR and NDCG, proving that user history plays an important role in our model structure. Without the user history, the model is only learning which items users frequently engage with, and thus has limited performance.

**(2). Item-by-item**. To answer **(Q2)**, we have tried a different version of our model, which concatenates the embeddings of each of the target items $\{m_{r_1}, m_{r_2}, \cdots m_{r_k}\}$ with the user history $(m_{s_1}, m_{s_2}, \cdots m_{s_n})$ individually as the input to the self-attention layer. Specifically, the input now becomes a batch of inputs (different from the minibatch in stochastic optimization).

$$\widehat{E}_{r_i} = \begin{bmatrix} E_{his} + P \\ m_{r,i} \end{bmatrix}, i = 1, 2, \cdots, k \tag{13}$$

Each of these concatenated inputs $\widehat{E}_{r_i}$ are feed into the self-attention layers one by one. The channel for the target item $r_i$ in the output is again split and feed into the MLP to predict the interaction score for the item $r_i$. In other words, $K$ forwards are needed for one set of target items $(r_1, r_2, \cdots, r_k)$. In this way, the transformer is capable of learning the interactions between the user history and the target items, but not the contrastive signals among the target items since they are not concatenated anymore. As can be observed, such a design performed worse than the original one, but it is still able to learn the interactions between the user history and the target items. Moreover, item-by-item training takes much more time than the vanilla design due to the excess number of forward passes.

**(3). BCE loss**. For **(Q3)**, we have used the binary cross entropy (BCE) loss to train our CTSRec model. It turns out that our model is quite robust to the choice of the loss function. When using the BCE loss, the model converges much slower than the baseline model, and achieves slightly higher accuracy. We emphasize that the obtained HR and NDCG results are still To align with the ReChorus library (Wang et al., 2020b) and its baselines, we have chosen to use the BPR loss throughout this paper for consistency except for this ablation study.

**(4). Separate transformers**. We have also tried a different neural network architecture to see the effectiveness of our concatenate-then-split (CTS) structure. Specifically, we have used one transformer on the user history embeddings and another transformer on the target items embeddings. The outputs of the two transformers are then multiplied using the dot product operation, similar to SASRec (Kang and McAuley, 2018) and TiSASRec (Li et al., 2020). The performance of this model is, however, worse compared to our CTSRec model because the transformers never consider all the embeddings together. This further proves our claim that it is important to learn both the interactions between the user history and the target items, and the contrastive signals among the target items.

**(5) - (8). # blocks** $B$. To understand how the model architecture affects the performance of CTSRec, we have tuned the number of transformer blocks in our model. As can be seen in Table 5, CTSRec behaves reasonably well with 3-5 transformer blocks.

**Number of Negative Samples.** To understand the effect of the number of negative samples on the different algorithms **(Q5)**, we have tuned the number on the three datasets and plotted the change in HR@5 with respect to the number of negative samples in Figure 2. We have chosen GRU4Rec, SASRec, TiSASRec, TimiRec, and ContraRec as the comparison baselines since they are the top-performers in Table 3 and 4 and many of them utilize self-attention in their model architecture. As can be observed, our model is the most sensitive model to the number of negative samples. As we increase the number of negative samples, our model quickly learns the contrastive signals among them and the ground truth to enhance the ultimate performance. Other models, on the other hand, either have limited ability to learn or completely ignore these information in the model design. For example, GRU4Rec has slightly better performance when we increase the number of negative samples. However, models such as SASRec, TiSASRec, and ContraRec are almost invariant to the number $k$.

## 5 Conclusions

In this paper, we propose a new transformer-based sequential recommendation model that explicitly incorporates the interactions between user history and target items, as well as the contrastive signals between the ground truth and negative items by a concatenate-then-split structure. Extensive experiments show that the new model achieves state-of-the-art performance on popular sequential recommendation tasks. Our ablation study also shows that these signals which are missing from prior works, are very important and helpful for

the sequential recommendation tasks. Future research directions include further combining our model with previous advanced techniques such as time-interval modeling (Li et al., 2020), multi-interest recommendation (Cen et al., 2020; Wang et al., 2022), and item relation modeling (Wang et al., 2020b;a).

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

# Appendix

## A    Experiment Details

In this appendix, we provide the experimental details and the additional experiments.

### A.1    Reproducibility

As we have mentioned in the Experiments section, we have used the ReChorus library as the codebase to develop our model and run the baseline models Wang et al. (2020b). For all the baseline models (e.g., GRU4Rec, SASRec, and ContraRec), we have used the default implementations and the default set of hyperparameters. The algorithms are trained for at most 200 epochs on the train set, evaluated on the dev set, and then the best model (in terms of NDCG@5 on the dev set) among all the models is used for testing on the testing set. In other words, all the results in Table 3 and 4 are evaluated on the test set with the best models of each algorithm. The training process is early stopped if the model's performance continues to deteriorate for 20 epochs. We have used 1e-3 as the learning rate for Adam when training CTSRec, which is the default value for the algorithm. The weight decay is set to be 1e-4. The batch size is set to be 256 (default in the ReChorus library).

### A.2    Running Time Comparison

In order to fairly compare performance and efficiency tradeoff  all the models, we have recorded the average epoch time of GRU4Rec, SASRec, TiSASRec, ContraRec, TimiRec, and CTSRec during the training stage. All experiments is conducted is based on AWS job queue using single A-100 GPU card. The implementation of all the baseline models are in the ReChorus library (Wang et al., 2020b). For a complete and fair comparison, we provide the epoch time for each of the algorithms in the cases $k = 2, 100, 200$ on the three Amazon datasets used in Table 3. As shown in Table 6, GRU4Rec converges the fastest with the smallest epoch time in all the cases. Our CTSRec model is the second fastest among all the models in the cases $k = 2$ and $k = 100$, and only slightly slower than TimiRec in the cases $k = 200$. Compared with the SOTA baselines, CTSRec outperforms them by a significant margin. At the same time, CTSRec is as fast as TimiRec, and much faster than ContraRec. To conclude, CTSRec is able to improve the performance of tranformer architecture with little performance cost overhead.

### A.3    Convergence Comparison

We further measure the speed of the algorithms based on convergence comparison. As shown in Table 7, the HR@5 of the algorithms on the dev set for the first 12 epochs when training all the algorithms on the Amazon G&GF dataset are recorded. As can be observed, CTSRec converges to its best performance within 10 epochs, where as the other SOTA algorithms such as ContraRec and TimiRec converge very slowly. In fact, ContraRec needs more than 100 epochs to achieve its best performance in Table 3 and 4, which is much slower than our CTSRec algorithm. If we combine the average epoch time information in Table 6 and the number of epochs needed to converge in Table 7, we can reach the conclusion that CTSRec is able to achieve its best performance in a small amount of time and epochs, and its performance is better than the SOTA methods.

Table 6: Average epoch time (seconds/s) comparison between the baseline methods and the new method CTSRec. The results are averaged over 10 independent runs. The lowest results in each column are highlighted in bold.

| Method / $k =$ | G & GF | | | CP&A | | | Games | | |
|---|---|---|---|---|---|---|---|---|---|
| | 2 | 100 | 200 | 2 | 100 | 200 | 2 | 100 | 200 |
| GRU4Rec | **9.9** | **12.7** | **16.1** | **10.1** | **13.4** | **16.7** | **11.9** | **16.8** | **21.6** |
| SASRec | 38.9 | 40.3 | 46.6 | 49.9 | 51.2 | 48.8 | 60.4 | 63.2 | 79.9 |
| TiSASRec | 45.7 | 50.3 | 55.9 | 53.3 | 56.6 | 64.5 | 66.4 | 77.6 | 92.3 |
| ContraRec | 153.1 | 159.4 | 172.8 | 163.7 | 171.3 | 183.0 | 225.5 | 245.2 | 254.6 |
| TimiRec | 20.3 | 26.3 | 33.7 | 20.2 | 29.0 | 34.9 | 28.6 | 39.3 | 49.2 |
| **CTSRec (ours)** | 14.3 | 20.1 | 36.7 | 14.1 | 21.8 | 37.8 | 18.1 | 29.7 | 50.4 |

Table 7: The evaluation HR@5 on the **dev** set for each algorithm after every epoch on Amazon G & GF dataset. The best performance in each row is highlighted in bold.

| Method / Epoch | 1 | 2 | 3 | 4 | 5 | 6 | 7 | 8 | 9 | 10 | 11 | 12 |
|---|---|---|---|---|---|---|---|---|---|---|---|---|
| FPMC | 0.258 | 0.327 | 0.375 | 0.401 | 0.409 | **0.411** | 0.411 | 0.406 | 0.402 | 0.397 | 0.393 | 0.388 |
| GRU4Rec | 0.336 | 0.392 | 0.405 | 0.422 | 0.431 | 0.435 | 0.443 | 0.454 | 0.459 | 0.459 | 0.460 | **0.464** |
| Caser | 0.338 | 0.379 | 0.390 | 0.397 | 0.412 | 0.422 | 0.431 | 0.446 | 0.451 | 0.449 | 0.451 | **0.456** |
| SASRec | 0.414 | **0.448** | 0.435 | 0.428 | 0.411 | 0.412 | 0.402 | 0.399 | 0.392 | 0.395 | 0.395 | 0.390 |
| TiSASRec | 0.252 | 0.323 | 0.360 | 0.381 | 0.403 | 0.414 | 0.427 | 0.434 | 0.438 | 0.441 | **0.444** | 0.443 |
| ComiRec | 0.249 | 0.283 | 0.322 | 0.349 | 0.366 | 0.377 | 0.385 | 0.391 | 0.399 | 0.403 | 0.408 | **0.412** |
| ContraRec | 0.077 | 0.090 | 0.114 | 0.144 | 0.177 | 0.203 | 0.223 | 0.240 | 0.255 | 0.269 | 0.278 | **0.288** |
| TimiRec | 0.252 | 0.330 | 0.356 | 0.381 | 0.407 | 0.426 | 0.443 | 0.453 | 0.461 | 0.468 | 0.471 | **0.472** |
| **CTSRec** | 0.383 | 0.377 | 0.404 | 0.428 | 0.449 | 0.465 | 0.463 | **0.471** | 0.463 | 0.459 | 0.458 | 0.451 |

