# OpenReview forum: "Concatenative Contrastive Sampling for Transformer-based Sequential Recommendation"
_TMLR — Rejected by TMLR_

### Review · Reviewer_3Za9 · 2023-11-21

**Summary Of Contributions:**

In the sequence recommendation modeling process of the Transformer, K candidate items are concatenated to the user's historical items. Subsequently, the model transforms the last K items in the sequence, producing the final scores.

**Audience:**

Yes

**Claims And Evidence:**

No

**Requested Changes:**

As metioned in Weakness part, clearly, this work has numerous issues, even including erroneous experimental metrics, making it unacceptable.

**Strengths And Weaknesses:**

1.Numerous works in sequence recommendation consider the target item as a guiding signal in modeling the sequence of historical items. However, this paper identifies such an approach as a flaw in existing methodologies, attributing it to insufficient research and survey.
2.The adoption of simple operations like inner product for calculating prediction scores is due to their simplicity and effectiveness. The method proposed in the paper is unacceptable complexity and impracticality in real-world applications, where the number of candidate items can reach up to 100 million.
3.The table format is irregular and lacks a definitive bottom line.
4.In the Causality section, the statement "Therefore, to prevent information leakage, the input embeddings for the items corresponding to (s1, s2, ..., st) should be masked when predicting the next item st+1" is obviously incorrect.
5.The paper frequently exhibits unclear expression of symbols. Such as k in Equation(11).
6.In the experimental section, the maximum length of the sequences is too short, typically requiring a length of 50 or more.
7.Using negative sampling as a testing method is biased while full ranking offers a more persuasive approach.
8.It is well-known that MovieLens is unsuitable as a dataset for sequence recommendation.
9.The expressions for the recommendation metrics given in Equation 12 are both incorrect.
10.As mentioned in Section 4.2, the method proposed in the paper does not demonstrate the capability to explicitly model information, which contradicts its ability to achieve better experimental results. Explicit modeling is essential for capturing users' sequential interests.
11.Setting a maximum limit of 200 epochs may prevent some baseline models from achieving full convergence.
12.The paper lacks an analysis of time complexity and space complexity, and theoretically, the proposed method is far more complex than the baselines.

---

### Review · Reviewer_P6Ud · 2023-11-27

**Summary Of Contributions:**

This paper is about sequential recommendation, one of the most fundamental recommendation tasks.
The authors proposed a concatenate-then-split structure that intentionally integrates historical interactions.
Results on two datasets show the the-of-the-art performance across prevalent sequential recommendation baselines.

Overall speaking, this paper is written well. However, the solution is a bit straightforward;  the baselines only include one method published in 2023; the utilized datasets are on a small scale. These weaknesses make the paper not suitable for publication in TMLR.

**Audience:**

Yes

**Claims And Evidence:**

Yes

**Requested Changes:**

1. The methodology.

- The method is not so well motivated. The authors claimed, "However, prior endeavors, while successfully leveraging user history attributes, are constrained in capturing the interplay between user history and new items, as well as the contrastive signals between authentic and unfavorable items." However, the statement is not well supported. I do not agree that all existing methods have such a limitation.
- The proposed method is a bit straightforward.


2. The experiments.
- The baselines only include one method published in 2023
- The utilized datasets are on a small scale.
- There are only two datasets: Amazon and ML-1M.

3. Others.
- Complexity analysis of the proposed method can be added.

**Strengths And Weaknesses:**

1. Sequential recommendation studied in this paper is very fundamental and important. Almost all recommendation engines in the world have deployed sequential recommendation algorithms.
2. The paper is written well.
3. The method is clearly presented.

---

> ### Author Response · Authors · 2023-12-15
> **Rebuttal to Reviewer P6Ud**
>
> Thank you for your reviews. We are sorry for the late response as the authors are very busy these days. We would like to address your questions as follows.
>
>
> 1. *The method is not so well motivated...I do not agree that all existing methods have such a limitation.*
>
> **Response:** Thank you! We will modify the sentence to make it less strong in the next version of our paper. However, according to our ablation experiments in Figure 2, most existing baselines are insensitive to the number of negative samples used, which means that few of them can catch the intrinsic information contained in the target items.
>
> 2. *The proposed method is a bit straightforward.*
>
> **Response:** We agree that our method is a bit straightforward. However, this is the small change that makes ordinary transformers outperform many existing strong baselines with complicated algorithm designs. We believe that simple methods that are effective are better than complex methods that are less effective. Moreover, we believe that novelty is not considered to be one of the acceptance criteria of TMLR, as mentioned here https://jmlr.org/tmlr/acceptance-criteria.html.
>
> 3. *The baselines only include one method published in 2023.*
>
> **Response:** We compare with these methods because they are the most related and the strongest baselines in sequential recommendation. If the reviewer believes there are other suitable baselines to compare with, we kindly ask the reviewer to provide them in the comments so we can add more experiments to the next version of our paper.
>
>
> 4. *The utilized datasets are on a small scale. There are only two datasets: Amazon and ML-1M*
>
> **Response:** We believe that Amazon and ML-1M are standard datasets in sequential recommendation tasks and there are many papers that use these datasets to evaluate the performance of sequential recommendation algorithms such as [1]-[3]. If the reviewer thinks there are other suitable datasets, we kindly ask the reviewer to provide their names so that we can add them to our experiments.
>
>
> 5. *Complexity analysis of the proposed method can be added.*
>
> **Response:** We will add the complexity analysis to the next version of our paper.
>
> [1]. Jiacheng Li, Yujie Wang, and Julian McAuley. Time interval aware self-attention for sequential recommendation. In Proceedings of the 13th International Conference on Web Search and Data Mining, WSDM ’20, page 322–330, 2020.
>
>
> [2]. Chenyang Wang, Weizhi Ma, Min Zhang, Chong Chen, Yiqun Liu, and Shaoping Ma. Toward dynamic user
> intention: Temporal evolutionary effects of item relations in sequential recommendation. ACM Transactions on Information Systems (TOIS), 39(2):1–33, 2020a.
>
>
> [3]. Chenyang Wang, Zhefan Wang, Yankai Liu, Yang Ge, Weizhi Ma, Min Zhang, Yiqun Liu, Junlan Feng, Chao Deng, and Shaoping Ma. Target interest distillation for multi-interest recommendation. In Proceedings of the 31st ACM International Conference on Information and Knowledge Management, CIKM ’22, page 2007–2016, 2022.

---

### Review · Reviewer_sN7K · 2023-11-30

**Summary Of Contributions:**

The paper proposes a new transformer-based architecture for sequential recommendation. Compared with traditional self-attention based sequential recommender systems, the paper proposes to concat the user histories and target item embeddings, then apply self-attention on top of it. The benefit is two-fold, one is to explicitly incorporate interactions between user history and target items; the other is utilize the contrastive signals between negative items and the ground-truth. Empirical performance shows better performance compared with other sequential-based recommendation models.

**Audience:**

Yes

**Claims And Evidence:**

Yes

**Requested Changes:**

See Weakness.

**Strengths And Weaknesses:**

Strength:

- The paper is well-motivated, well-written and easy to read.
- The designed new architecture is simple, easy to implement, and shows strong empirical performance.
- Ablation studies are performed to examine different aspects of the method, such as the number of negative samples, the training loss, the cut of user-history, etc.

Weakness:
- All the empirical results only provided the average, could we also report the std to understand the significance of the gain?
- Some of the design choice are not super clearly explained, for example, the cut-off of the user-history output in the final prediction, there are other ways to match the shape of the labels as well, for example, learning a weight matrix to get k rows, etc. It would be great if the authors could explain in details about the design choices here.
- The additional computational complexity is not discussed.
- Did the authors try a baseline approach where utilize two transformers separately for both the user history and targeted items? This might be able to provide an examination of the relative effect of contrastive signals.
- This seems a purely empirical paper on recommender systems, and I am not sure whether it fits the general audience of TMLR.

---

> ### Author Response · Authors · 2023-12-15
>
> Thank you for your reviews. We apologize for the late response as the authors are very busy these days. We would like to address your questions as follows.
>
> 1. *All the empirical results only provided the average, could we also report the std to understand the significance of the gain?*
>
> **Response:** Thank you and we will try our best to run all the experiments within these weeks and add standard deviations to the next version of our paper. However, we also want to remark that the standard deviations are very small in sequential recommendations and very often not reported.
>
>
> 2. *Some of the design choice are not super clearly explained ... if the authors could explain in details about the design choices here.*
>
> **Response:** The design here is merely the simplest design because those columns initially contains the embeddings of the target items. We agree that a matrix projection is also a valid way to match the dimensions, but we did not want to add more complexities to the design as the transformer already introduces some computational overhead. We will add more discussions in the next version of our paper.
>
>
>
> 3. *The additional computational complexity is not discussed.*
>
> **Response:** Thank you for bringing it up. We will add the discussion of computational complexity to the next version of our paper.
>
> 4. *Did the authors try a baseline approach ... relative effect of contrastive signals.*
>
> **Response:** That is a great idea for ablation study! Thank you for bringing it up. We will try our best to add this ablation study to the next version of our paper.
>
>
> 5. *This seems a purely empirical paper on recommender systems, and I am not sure whether it fits the general audience of TMLR.*
>
> **Response:**  We believe that TMLR welcomes papers with only empirical results. Please find the submission guidelines here https://jmlr.org/tmlr/editorial-policies.html.

---

### Decision · Action_Editor_25dy · 2024-01-26

**Recommendation:** Reject

**Comment:**

The reviewers' decision came back split (1 weak accept, 1 weak reject, and 1 strong reject).

Below, I summarize the main elements that the reviewers used to motivate their decision and provide my assessment.

1. **Novelty and significance of contribution.** While reviewers raised this, this aspect should not be considered per TMLR's guidelines.


2. **TMLR Audience.** Reviewers argue that perhaps TMLR is not the right audience for this paper. According to TMLR's guidelines, it suffices that some of the TMLR audience would be interested in the findings of this paper. This paper provides a simple modification to the current sequential recommendation architecture that seems to improve performance. As such, I think it would be of interest, particularly to those interested in recommender systems.


3. **Motivation, claims, and evidence.** Here, the reviewers mention that the authors tried to improve their text (in the most recent version of the manuscript) but that some of the claims are still not well supported. While I believe that the authors provide some evidence of their results (including the results in Figure 2 and an ablation study), I also find that the paper could more clearly support its claims and improve its motivation with some additional modifications, including, for example:
    1. A more precise description of your research question and the limitations of previous work (currently, the motivation is somewhat informally described in the introduction). In particular, it would be helpful to describe the information from the target items you are attempting to model and how it might be useful for the recommendations. Further, describing the expected effect of the size of the negative set would be interesting. Perhaps using a (made-up) example would suit that.
    2. More evidence that your proposed contribution captures this extra information (in addition to Figure 2 and the ablation study). For example, probing the resulting model and/or providing an exploratory analysis could help.
    3. The results are shown on two datasets. Several other datasets have been used in previous papers (including the ones you've cited and even different Amazon categories) that would serve to solidify current evidence and perhaps provide a slightly more in-depth analysis.

   Again, these are just examples of elements you could consider adding to your paper to clarify some claims and evidence.


4. **Missing recently proposed baselines.**  Here, unfortunately, the reviewer did not make concrete suggestions. On one end, it is possible (maybe even likely) that new baselines appeared in the last year. I suggest the authors carefully review the recent literature for relevant sequential recommendation methods. (To be clear, this was not a motivation for suggesting the rejection.)


In short, my assessment is that this paper is closer to being acceptable at TMLR than the reviewers' decision might lead one to believe. This is partly because the reviewers have used criteria from other venues (such as novelty and significance). With that in mind, I still find that 1) further evidence should support the claims, and 2) the paper's motivation should be described more precisely.

I would happily consider a revised version of this manuscript.

**Audience:**

Yes, recommender systems are, amongst other things, a valuable application for ML models. While there exist specialized recommender systems journals and conferences, it is not unusual for recommender systems' papers to appear in more general ML conferences. As such, these findings would interest some individuals in TMLR's audience.

**Claims And Evidence:**

This was an element raised by reviewers both as part of their initial reviews (and also in their final recommendations). The authors did provide an updated version of their paper meant to clarify and weaken some of their claims. However, it's the reviewer's opinion that the current version of the manuscript still requires improvements—more in my comments below.

**Resubmission Of Major Revision:**

The authors may consider submitting a major revision at a later time.